# Electrical tuning of branched flow of light

Shan-shan Chang[1,9], Ke-Hui Wu[2,9], Si-jia Liu[3], Zhi-Kang Lin[4], Jin-bing Wu[3], Shi-jun Ge[3], Lu-Jian Chen [2], Peng Chen [3], Wei Hu [3], Yadong Xu [4], Huanyang Chen [5], Dahai He[5], Da-Quan Yang[6], Jian-Hua Jiang [4,7] ✉, Yan-qing Lu [3] ✉ & Jin-hui Chen [1,8] ✉

Branched flows occur ubiquitously in various wave systems, when the propagating waves encounter weak correlated scattering potentials. Here we report the experimental realization of electrical tuning of the branched flow of light using a nematic liquid crystal (NLC) system. We create the physical realization of the weakly correlated disordered potentials of light via the inhomogeneous orientations of the NLC. We demonstrate that the branched flow of light can be switched on and off as well as tuned continuously through the electro-optical properties of NLC film. We further show that the branched flow can be manipulated by the polarization of the incident light due to the optical anisotropy of the NLC film. The nature of the branched flow of light is revealed via the unconventional intensity statistics and the rapid fidelity decay along the light propagation. Our study unveils an excellent platform for the tuning of the branched flow of light which creates a testbed for fundamental physics and offers a new way for steering light.

When waves propagate through a medium with weakly correlated disorder potentials, they will split into randomly focused, branched transmission channels, leading to strong fluctuations in wave intensity —a fundamental phenomenon known as branched flows. These branched transmission channels originate from the random caustics in wave propagation and are a ubiquitous effect taking place in wave dynamics in the realm between the ballistic and diffusive limits. Since the observation of branched flows in two-dimensional electron gas[1–6], this phenomenon has been extended to various physical systems, such as electromagnetic waves[7,8], sound waves[9], elastic waves[10], and water waves[11,12]. In nonlinear wave dynamics, branched flows can also act as an activation mechanism for the appearance of extreme wave events such as tsunamis waves and rogue waves[11,13–15]. Recently, a branched flow of light was discovered with non-diffracting filaments when a coherent laser beam passes through a thin soap membrane[16,17]. The

formation of branched flows is mainly attributed to the wave scattering from correlated weak disorder potentials, under which waves form concentrated ray caustics with high wave intensities. This fundamental phenomenon has been discussed extensively in theory[14,18–25]. However, owing to their erratic nature and rich behaviors[2,26], manipulation of branched flows in a controllable manner has never been realized in experiments, although it is highly anticipated, especially as a new perspective in steering light and other waves[27–30].

Here we demonstrate the electrical tuning of the branched flow of light in a nematic liquid crystal (NLC) film using an experimental setup illustrated in Fig. 1a. The liquid crystal film naturally hosts lots of topological defects such as disclinations and boojums. These topological defects contribute to the spontaneous formation of structured patterns with disordered orientations of molecules within the film plane (called schlieren textures). Such disordered orientations lead to

[1]Institute of Electromagnetics and Acoustics, Xiamen University, Xiamen 361005, China. [2]Department of Electronic Engineering, Xiamen University, Xiamen 361005, China. [3]College of Engineering and Applied Sciences, Nanjing University, Nanjing 210023, China. [4]School of Physical Science and Technology & Collaborative Innovation Center of Suzhou Nano Science and Technology, Soochow University, Suzhou 215006, China. [5]Department of Physics, Xiamen University, Xiamen 361005, China. [6]State Key Laboratory of Information Photonics and Optical Communications, School of Information and Communication Engineering, Beijing University of Posts and Telecommunications, Beijing 100876, China. [7]Suzhou Institute for Advanced Research, University of Science and Technology of China, Suzhou 215123, China. [8]Innovation Laboratory for Sciences and Technologies of Energy Materials of Fujian Province (IKKEM), Xiamen 361005, China. [9]These authors contributed equally: Shan-shan Chang, Ke-Hui Wu. ✉e-mail: jianhuajiang@suda.edu.cn; yqlu@nju.edu.cn; jimchen@xmu.edu.cn

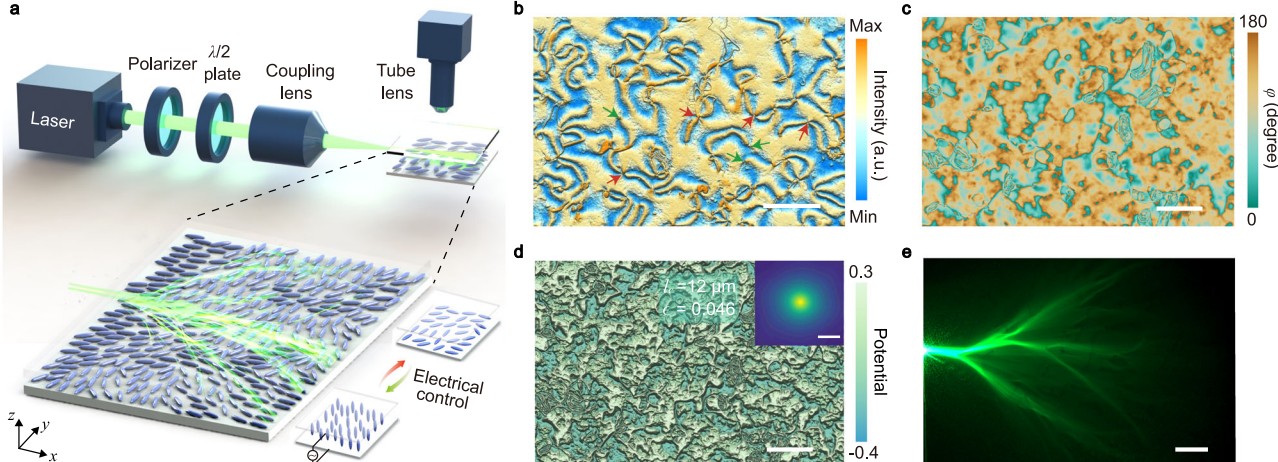

**Fig. 1 | Branched flow of light in a disordered nematic liquid crystal (NLC) film.**
**a** Experimental setup. The inset shows the schematic of the branched flow of light in an NLC film with a focused Gaussian beam input. The NLC molecules are randomly oriented in a glass cell. The arrangement of the NLC molecules is tunable via the electrical voltage bias. **b** Schlieren textures of NLC film (without electrical voltage) observed under a cross-polarized microscope. The green and red arrows indicate disclinations of 1/2 and 1 topological charges, respectively. The scale bar is 200 μm. **c** Measured distribution of the in-plane angle $\varphi$ of the director at zero electrical voltage. The scale bar is 200 μm. **d** Reconstructed effective optical potential from the NLC director distribution in **c**. The scale bar is 200 μm. The inset shows the autocorrelation function, correlation length $l_c$, and strength $\epsilon$ of the effective optical potential. The scale bar in the inset is 20 μm. **c**, **d** are for the same region, while **b** is from a different region. **e** Observed typical landscapes of branched flow of light showing self-similar features. The scale bar is 200 μm. The thickness of the NLC film is 20 μm.

inhomogeneous dielectric anisotropy in the NLC film which acts as a weakly disordered potential for the propagating light. This disordered potential, whose correlation length is larger than the wavelength of the propagating light, can be further tuned via the electro-optical properties of the NLC molecules. With a small electrical voltage, the disordered potential for photons can be suppressed by aligning the NLC molecules along the electrical field. Remarkably, due to the stability of the topological defects under the electrical field, this tuning process is reversible and repeatable. The branched flow can be switched on and off, and the branching properties can be tuned quantitatively through the electrical voltage bias. Furthermore, thanks to the planar optical anisotropy, the disordered potentials for photons can also be manipulated by the polarization of the incident light, providing another way for the quantitative tuning of the branched flow of light.

## Results
### Theory of branched flow of light in liquid crystal film
According to the Ericksen-Leslie theory, the optical properties of liquid crystals can be characterized by the spatial distribution of molecules' orientation vector (or the "director") $\hat{\mathbf{n}}(\mathbf{r})$[31,32]. This unit vector determines the local average molecular orientation. Due to the optical birefringence, a uniform NLC system supports light propagation with two different polarizations, i.e., ordinary and extraordinary waves. The ordinary (extraordinary) wave has the polarization perpendicular (parallel) to the plane spanned by the wavevector of the light and the director $\hat{\mathbf{n}}$. As a consequence, the ordinary-wave has a constant refractive index, which does not depend on the orientation of the molecules. In contrast, the refractive index of the extraordinary wave depends on the director $\hat{\mathbf{n}}$. In our experimental setup, the effective refractive index for the incident light with polarization along the $y$-direction has the following dependence on the director orientation lying in the film plane (Supplementary Fig. S2, see Supplementary Note 1).

$$n_{\text{eff}}(\mathbf{r}) = \frac{n_\perp n_\parallel}{\sqrt{n_\perp^2 \sin^2\varphi + n_\parallel^2 \cos^2\varphi}}, \qquad (1)$$

where $\varphi$ is the in-plane azimuthal angle of the local director $\hat{\mathbf{n}}(\mathbf{r})$, $n_\perp$ and $n_\parallel$ are ordinary refractive index and extraordinary refractive index,

respectively. In an NLC cell without special treatment, the molecules' orientations are not uniform but disordered. These molecules form schlieren textures with various topological defects, which are generated by the surface forces induced by the roughness of the interfaces between the NLC film and the cladding glass substrate[33]. In experiments, the schlieren textures can be observed directly using a cross-polarized optical microscope, as shown in Fig. 1b. These topological defects (as indicated by the arrows in Fig. 1b) lead to the spontaneous formation of the disordered orientations of the NLC molecules.

The propagation of the extraordinary wave in the NLC film is described by the following two-dimensional scalar Helmholtz equation within the small-angle scattering approximation (see Supplementary Note 1)[29,34]:

$$-\nabla_\perp^2 \psi + k_0^2(\bar{n}^2 - n_{\text{eff}}^2(\mathbf{r}))\psi = k_0^2 \bar{n}^2 \psi. \qquad (2)$$

Here, $\nabla_\perp^2 = \frac{\partial^2}{\partial x^2} + \frac{\partial^2}{\partial y^2}$, $\psi$ is the amplitude of the extraordinary wave, and $k_0$ is the wavevector of the light in vacuum. $\bar{n}^2 \equiv \langle n_{\text{eff}}^2(\mathbf{r})\rangle$ is the square of the effective refractive index averaged over the whole observation area. It is crucial to note that Equation (2) resembles the time-independent Schrödinger equation where the total energy is $E_{\text{tot}} = k_0^2 \bar{n}^2$ and the effective potential energy is $V(\mathbf{r}) = k_0^2(\bar{n}^2 - n_{\text{eff}}^2(\mathbf{r}))$. The latter acts as the scattering potential with a zero average value, $\langle V(\mathbf{r})\rangle = 0$. Here, since the NLC film's thickness is much larger than the wavelength of the light, the effective index of the waveguide mode is very close to the liquid crystal index (Supplementary Fig. S3), while the waveguiding-induced effective index change can be neglected. We remark that the ordinary wave, in contrast, has a constant refractive index and thus does not suffer disordered potential nor support branched flows.

By measuring the spatial distribution of the director's azimuth angle via polarization interference microscopy (see Fig. 1c and Methods), the potential energy can be obtained based on Eqs. (1) and (2) (see Fig. 1d). Although branched flows have quite complex patterns, their main properties are determined by the total energy $E_{\text{tot}}$ and the correlation of the disordered potential $c(\mathbf{r}) = \langle V(\mathbf{r}+\mathbf{r}')V(\mathbf{r}')\rangle = 4\epsilon^2 E_{\text{tot}}^2 f(\mathbf{r}\cdot\mathbf{r}/l_c^2)$. Here,

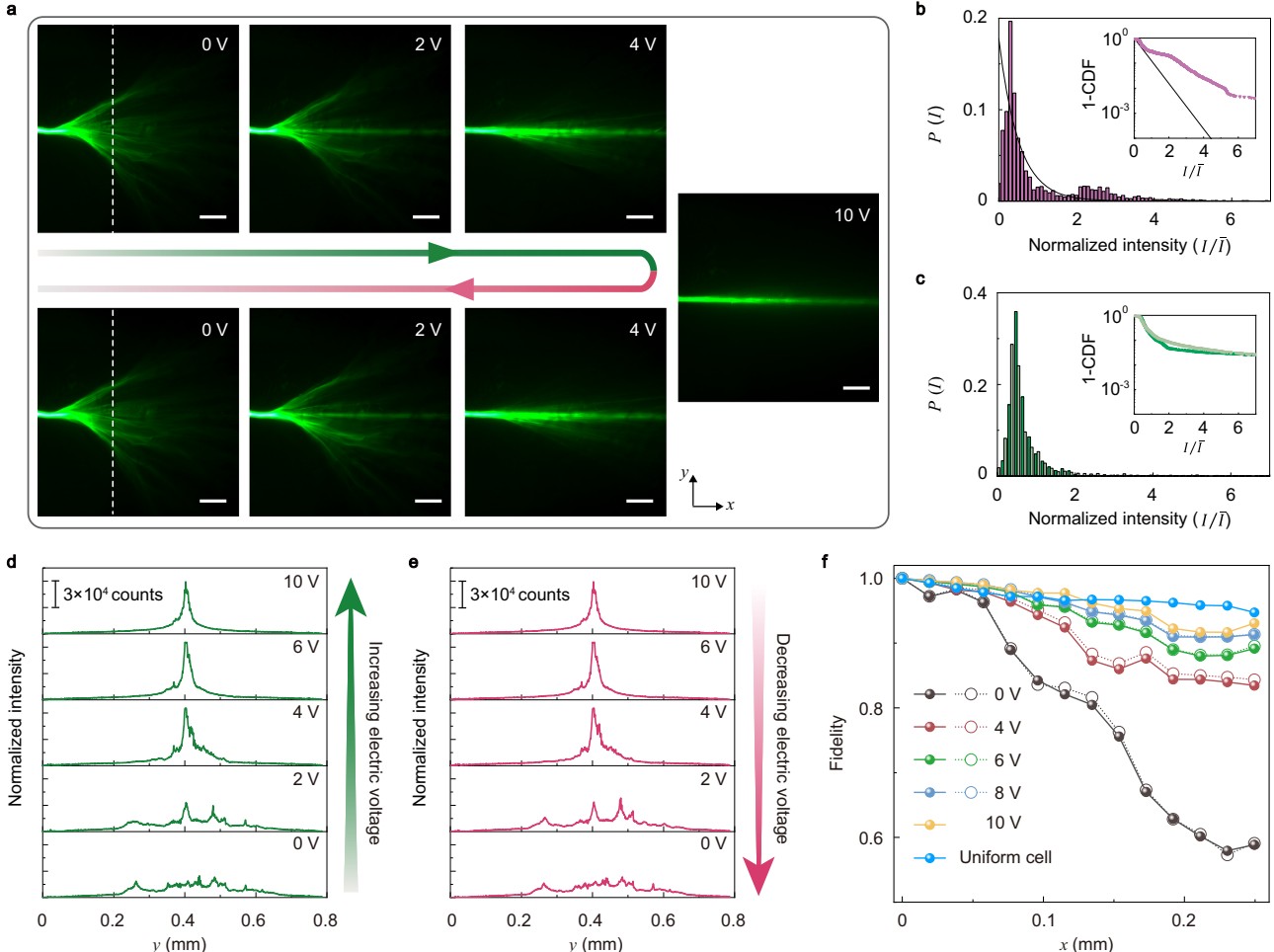

**Fig. 2 | Electrical tuning of the branched flow of light. a** Electrical switch off and on of the branched flow of light. The scale bar is 100 μm. **b** Statistical distribution of the optical intensity (purple columns) at the white-dashed line in **a** without electrical voltage. The black curve represents a Rayleigh distribution. The inset shows the cumulative distribution function (CDF) of the optical intensity in the semi-log axis. The black curve represents again the Rayleigh distribution. **c** Probability distributions of the optical intensity with an electrical voltage bias of 10 V at the incident (olive green columns) and at the white-dashed line in **a** (greyish green columns). Inset: CDF of the two distributions of the optical intensity. **d**, **e** The single-shot traces of cross-section optical intensity at the white-dashed line in **a** during the processes of increasing electrical voltage (**d**) and decreasing electrical voltage (**e**). **f** Fidelity of the propagating optical field versus the propagation distance for various electrical voltages. The solid (hollow) symbols indicate the measured results for the process of increasing (decreasing) electrical voltage. The fidelity of propagating fields in a uniform-alignment NLC cell is used as a benchmark. The thickness of the NLC film is 10 μm.

$\epsilon = \sqrt{c(0)}/2E_{tot} = \sqrt{\langle V^2 \rangle}/2E_{tot}$ characterizes the relative strength of the optical potential, and $f(x)$ is a power-law decay function with $f(0) = 1$ (Supplementary Note 1). In branched flows, the main features are not sensitive to the details of the optical potential but depend on the correlation length $l_c$ and the relative strength $\epsilon$ of the random potentials[16]. At zero electrical voltage bias, we find from the results in Fig. 1d that in our system $\epsilon = 0.046$ and $l_c \simeq 12$ μm. Therefore, the scattering potential is quite weak and its correlation length is much larger than the wavelength of the light. These two properties provide the foundation for the small-angle scattering approximation in Eq. (2) and the emergence of a branched flow of light.

## Electrical tuning of the branched flow of light

In the experimental setup, the NLC (E7) layer is confined by capillarity in a planar cell. Two treated glass slides coated with indium-tin-oxide (ITO) which enables the application of a low-frequency (1 kHz) voltage bias, serve as the cladding for the NLC (see Methods). In this work, all the gate voltages refer to the peak-to-peak values ($V_{pp}$) in a square waveform by default. The E7-NLC medium shows high optical

birefringence: $n_\perp = 1.52$, $n_\parallel = 1.74$ in the visible spectrum at room temperature[35]. A laser light of wavelength 532 nm and linear polarization along the $y$-direction is coupled into the NLC film and propagates along the $x$-direction. Due to the intrinsic light scattering in the NLC film, the landscapes of the branched flow of light can be directly observed by an optical microscope system with a digital camera. The propagating light continues to branch off into even smaller tributaries, showing an intriguing, complex, self-similar branching geometry (see Fig. 1e).

Since the orientations of the NLC molecules can be modulated by the electrical voltage bias $V_{pp}$, the effective optical potential $V(\mathbf{r})$ and hence the light scattering can be tuned electrically. Figure 2a shows the electrical tuning of the light scattering in the NLC film. At zero voltage bias, the optical intensity distribution shows clear features of the branched flow of light: The light beam splits into different paths after propagating about 100 μm. With the increase of electrical voltage bias, the branched flow of light with strong intensity fluctuations in a large area is gradually suppressed. At $V_{pp} = 2$ V, an unperturbed branch propagating straightly along the $x$-direction emerges. With $V_{pp} = 4$ V, this unperturbed branch is enhanced and becomes the mainstream, while the other branching flows are reduced. At $V_{pp} = 10$ V, the

unperturbed branch becomes dominant and the other branching flows are suppressed, leading to a beam propagation experiencing natural diffraction with increasing width and decreasing on-axis intensity. This evolution process can be understood as due to two main effects. First, the alignment of the NLC molecules toward the $z$-direction by the electrical voltage makes the orientation of the NLC molecules more uniform and thus suppresses the fluctuation of the effective optical potential. This scenario strongly reduces the light scattering and thus the branched flows. Second, the alignment of the NLC molecules also tilts the optical axes of the NLC film, making the ordinary wave component increase and the extraordinary wave component decrease from the birefringent effect. Since the ordinary wave component does not experience the random effective optical potential nor support the branched flow of light, an unperturbed light flow emerges and its intensity increases with the electrical voltage bias. In fact, the branched flow pattern can be tuned by the polarization of the incident light (Supplementary Fig. S18), as shown and discussed later.

A key signature of the branched flow is its unconventional intensity statistics. By analyzing the optical intensity distribution at the dashed line in Fig. 2a, we obtain the distribution and its cumulative distribution function (CDF). In conventional light scattering, such a distribution will be a Rayleigh distribution: $P(I) = e^{-I}$, where $I$ is the normalized field intensity[7]. Here, due to the complex branched flows, the optical intensity distribution has a notable bump at large intensities (Fig. 2b), reflecting the emergent high-intensity branches with the concentrated ray caustics[14,18,19]. In the CDF (see Fig. 2b inset), this feature is manifested as the long tails at a high optical intensity region (larger than the average intensity) that deviates significantly from the Rayleigh distribution. With an electrical voltage bias of $V_{pp} = 10$ V, the light beam simply experiences natural diffraction with an optical intensity distribution close to the input light (Fig. 2c), showing features of a nearly scattering-free beam.

Remarkably, although the branched flows have quite complex geometries, in our system the electrical tuning of them is highly repeatable and reversible. As shown in Figs. 2d, e, the optical field distribution at the dashed line in Fig. 2a can be recovered when we first increase and then decrease the electrical voltage. We believe that this reversibility is contributed by the schlieren textures of NLC film which are modified but not destroyed by the electrical voltage, probably due to the robustness of topological defects as pinned by the surface forces which are induced by the roughness of the interfaces between the NLC film and the cladding glass substrate (Supplementary Fig. S7)[32]. In addition, with the electrical voltage varying from 0 to 10 V, the width of the optical field distribution changes from ~ 0.42 mm to ~ 0.02 mm. This substantial change (~21 times) and its continuous tunability are a significant aspect of our system. We note that the branched flow field patterns can also be tuned by temperature control as the birefringence vanishes when the temperature is approaching the phase transition point of NLC (Supplementary Fig. S20a). However, we observe that the thermal tuning method is highly irreversible for the branched photonic patterns (Supplementary Fig. S20b–c). This is probably due to the collision of the growing isotropic-nematic interface during the cooling process, which can result in a higher energy metastable state of the NLC and complex changes in the Schlieren textures[36].

To quantitatively characterize the electrical tuning of the light scattering and the branched flows, we use fidelity to describe the deformation of the optical field along the propagation direction (i.e., the $x$-direction),

$$F(x) = \frac{\int dy I(x_0, y) I(x, y)}{\sqrt{(\int dy I^2(x_0, y))(\int dy I^2(x, y))}}, \qquad (3)$$

where $I(x_0, y)$ and $I(x, y)$ are the cross-section intensity distribution at the starting position ($x_0$) and at a later position ($x > x_0$) along the

propagation direction, respectively. In a scattering-free system, the fidelity decays slowly along the propagation direction. Thus the fidelity quantitatively characterizes the light scattering during the propagation. From Fig. 2f, one finds that at $V_{pp} = 10$ V, the fidelity indeed decays slowly along the $x$-direction, as light scattering is suppressed by the electrical voltage. Specifically, the fidelity remains as large as 0.92 even after light propagation for a distance of 0.3 mm. To further nail down the effect of the gate voltage, we also fabricate a uniform-alignment NLC cell (see Methods) as a benchmark. In the uniform NLC cell, the light propagation only shows natural diffraction features (Supplementary Fig. S11). The measured fidelity curve for the uniform NLC cell (the blue curve in Fig. 2f) shows a slow decay behavior similar to that of the disordered sample with $V_{pp} = 10$ V. At small gate voltages, the fidelity decays much faster, signaling the emergence of the branched flows.

The fidelity also shows the reversibility of the NLC system under increasing and decreasing electrical voltages. At zero electrical voltage, the fidelity decays rapidly along the $x$-direction, indicating the branched flow of light. Interestingly, here the fidelity does not decay uniformly as in random walk, but decays in a step-like form with each step close to the branching distance $d_f$, indicating that branched flow has a significant contribution in the fidelity decay as it leads to strong fluctuation in the optical field. In this work, we estimate the branching distance as $d_f \simeq 0.13$ mm which, as shown later, is consistent with the analysis from various experimental and theoretical aspects. For instance, a rough theoretical estimation based on the disordered optical potential gives $d_f \simeq 0.1$ mm which is not far away from the experimental observation (Supplementary Note 1 and Supplementary Fig. S15).

To characterize the emergence of the first branch, we investigate the branched flow of light with a quasi-plane-wave incident. The quasi-plane-wave light is approximated by a broad elliptic beam generated by a cylindrical lens[16] (Supplementary Fig. S8 and Methods). As shown in Fig. 3a, b, starting from a uniform optical field, the incident light gradually evolves into random caustics with strong intensity. These random caustics emerge at a propagation distance $\simeq d_f$, i.e., the first branching distance. Due to the considerable non-dissipative propagation loss from the disordered director distribution of NLC film, for example, light scattering due to long-range collective orientation fluctuations of the molecular axis of NLC[37] and the limited resolution of the instrument, only the first branching can be clearly observed (Supplementary Fig. S6). We measure the scintillation index, i.e., the normalized variance of the optical intensity distribution[8,16],

$$S(x) = \frac{\langle I^2(x, y) \rangle}{\langle I(x, y) \rangle^2} - 1 \qquad (4)$$

along the propagation direction, where the average is taken over the transverse $y$-coordinate and over different measurements (Supplementary Fig. S5). From the scintillation index in Fig. 3c, one finds that the variance of the optical field is maximized at $x \simeq 0.15$ mm. This maximal fluctuation in the optical field is coincident with the formation of strong, random caustics due to the weakly correlated light scattering and thus indicates the emergence of branching behavior. Therefore, the branching distance here is $d_f \simeq 0.15$ mm, which is close to the estimation from the fidelity.

With the increase of the electrical voltage, the scattering potentials and the branching behaviors are suppressed as the NLC molecules are aligned toward the $z$-direction. In the scintillation index, this is manifested in the decrease of the scintillation contrast (SC), i.e., the difference between the peak and the baseline of a scintillation curve. As shown in Fig. 3c, d, the peak scintillation index, and the SC are considerably reduced with the gating electrical voltage. A systematic investigation (Fig. 3d) shows that the SC decreases continuously with the electrical voltage while a notable reduction starts from $V_{pp} = 1.6$ V.

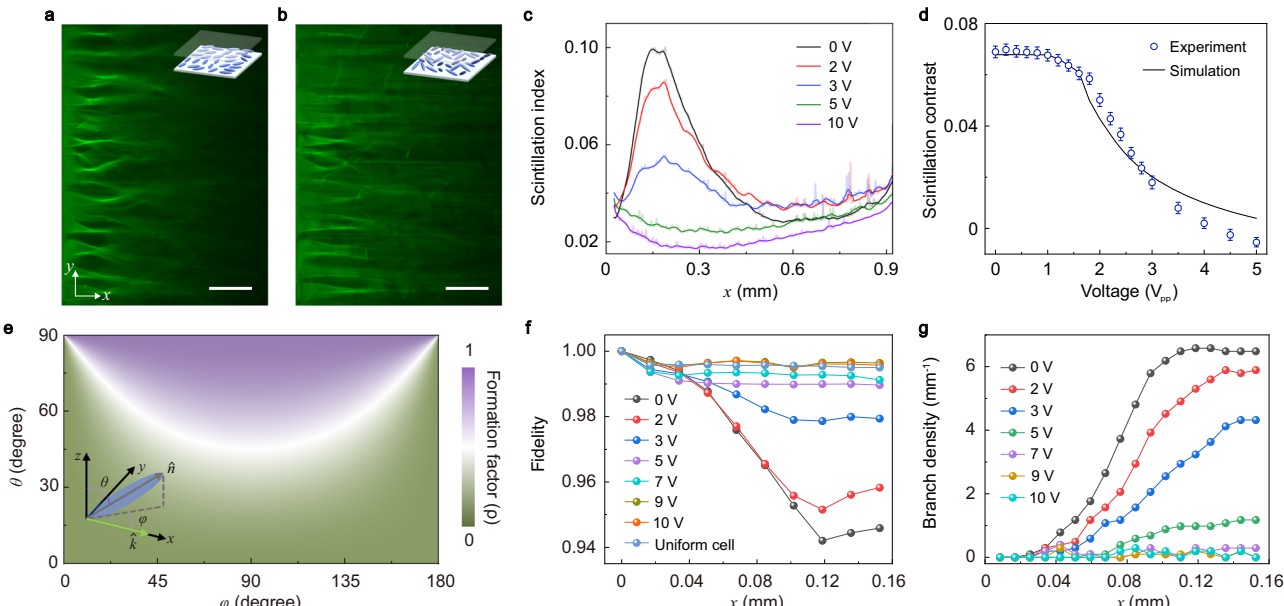

**Fig. 3 | Electrical tuning of the branched flow of light with quasi-plane-wave incident. a, b** Measured profiles of the optical field at different electrical voltages ($V_{pp}$): (**a**) 0 V and (**b**) 3 V. The incident light is propagating along the $x$-direction with a linear polarization along the y-direction. Insets: Schematic diagram of the NLC director distributions. The scale bar is 200 μm. **c** The scintillation index of the optical field along the propagation direction (averaged over ten repetitions) with various electrical voltages. **d** Measured and simulated scintillation contrast versus the electrical voltages. The error bar represents the standard deviation.

**e** Dependence of the formation factor $\rho$ on the azimuth angle $\varphi$ and the polar angle $\theta$ of the NLC director. Inset: Schematic of the NLC director $\hat{n}$ and the optical wavevector **k** along the $x$-direction. **f** Measured optical field fidelity versus the propagation distance along the $x$-direction for various electrical voltages. The fidelity of a uniform-alignment NLC cell is used as a benchmark. **g** Measured average branched density (averaged over four realizations) versus the propagation distance for various electrical voltages. The thickness of the NLC film is 20 μm.

Moreover, we find that the observed transition voltage is close to the calculated Fréedericksz transition threshold $\simeq 1.8$ V (Supplementary Note 3), which clarifies the electrical reorientation of the liquid crystal molecules as the underlying mechanism for the electrical tuning of the branched flow.

As the NLC molecules are aligned toward the $z$-direction by the electrical voltage, the extraordinary wave component of the incident light decreases while the ordinary wave component increases. Since the ordinary wave does not experience random scattering, the branched flows are suppressed, and the unperturbed light propagation increases. This scenario, together with the suppression of the random potential experienced by the extraordinary wave, reduces the SC when the electrical voltage increases. To quantify the portion of the extraordinary wave in the incident light that has a linear polarization characterized by the angle $\alpha$, we introduce the local formation factor which describes the intensity percentage of the extraordinary wave in the light field (see more details in Supplementary Note 1).

$$\rho = \frac{(\sin\theta\sin\varphi\cos\alpha + \cos\theta\sin\alpha)^2}{\cos^2\theta + \sin^2\theta\sin^2\varphi}. \tag{5}$$

As shown in Fig. 3e, as $\theta$ decreases from 90° to 0° (driven by the increasing electrical voltage), the $\rho$-factor of y-polarized light ($\alpha = 0°$) decreases from 1 to 0. The electrical voltage-induced change of the $\rho$-factor can be used to quantitatively estimate the statistical properties of the branched flow of light. We simulate the field-induced orientation in an NLC film with increasing electrical voltage (see Methods and Supplementary Fig. S17) and calculate the voltage-dependent of the SC. The simulation agrees well with the experimental results in Fig. 3d. We note that the gate voltages mainly manipulate the optical potential strength by controlling the out-of-plane orientation angle of the liquid crystal director, whereas the correlation length of the optical potential is not considerably affected (Supplementary Fig. S15). Although the branching distance

can be modified by the gate voltage, it is very difficult to directly observe this effect because of the considerable loss in light propagation and the inevitable excitation of the ordinary-wave component.

We also analyze the branching behavior via the fidelity of the optical field and the branch density for the quasi-plane-wave incident light. As shown in Fig. 3f, the fidelity decay along the propagation direction is suppressed by the increasing electrical voltage. These results also indicate that at $V_{pp} = 0$, the rapid decay of the fidelity takes place with a propagation distance of 0.12 mm which is close to the branching distance $d_f$ obtained from the results in Fig. 2f and Fig. 3c. The fidelity of propagating fields in the uniform NLC cell is nearly the same as that of the high gating-voltage ($V_{pp} = 10$ V) disordered sample, as expected.

The average number of caustics per unit length (i.e., the branch density) is another important statistical property of the branched flow[19]. This quantity, though plays an important role in understanding branched flows[18,19,38], has not yet been examined in experiments. Here we measure the branch density of the incident light versus the propagation distance as shown in Fig. 3g. The branch density increases quickly with the propagation distance and soon reaches a plateau, which agrees with the short-time asymptotic statistical behaviors of branched flows as predicted in Ref. 19 (see more details in Supplementary Note 4). With the increase of the applied voltage, the branch density decreases and eventually vanishes at a large voltage $V_{pp} = 10$ V due to the suppression of the random scattering potentials.

### Polarization-dependent branched flow of light
As the ordinary wave does not feel the spatial fluctuation of the director $\hat{n}$ nor does it support branched flows, we can also tune the branched flows via the polarization of the incident light at zero electrical voltage (see Fig. 4a-c, Supplementary Fig. S19). Such tuning is a direct manifestation of that the branched flows are determined by the extraordinary wave. Here the linear polarization of the incident light is

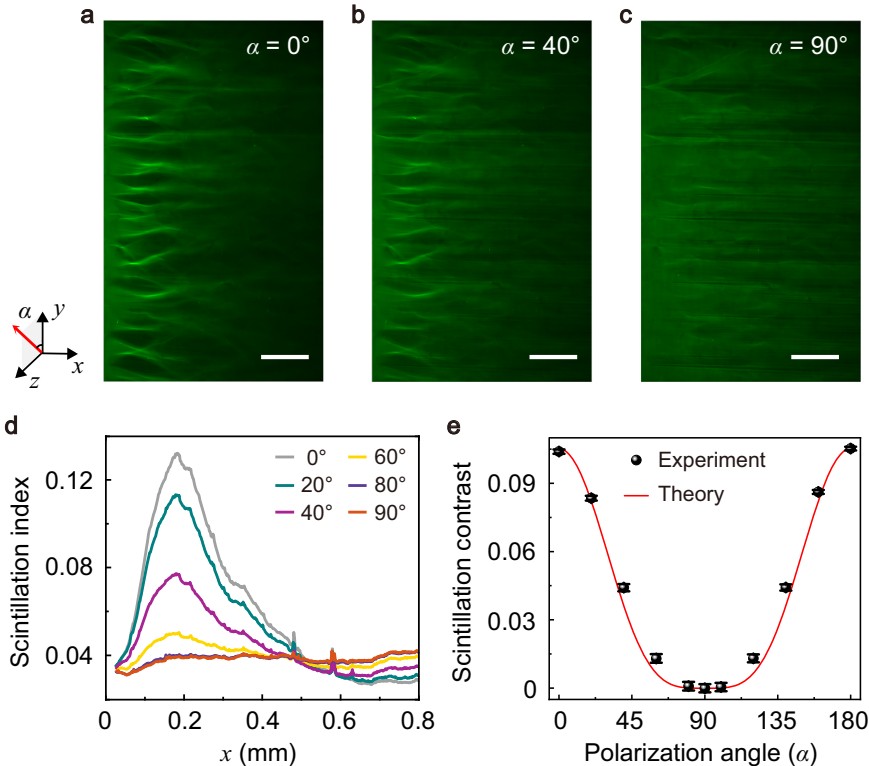

**Fig. 4 | Polarization control of the branched flow of light. a–c** Measured profiles of the optical field with the linearly-polarized, quasi-plane-wave light incident: **a** polarization angle $\alpha = 0°$, **b** $\alpha = 40°$, and **c** $\alpha = 90°$. The scale bar is 200 μm. The inset shows the linear polarization ($y$-$z$ plane) angle $\alpha$ of the incident light with respect to the coordinate system. **d** Extracted scintillation index of the optical field versus the propagation distance (averaged over six realizations) with various polarization angles $\alpha$. **e** Measured scintillation contrast and a theoretical fit with $\cos^4 \alpha$ for various polarization angle $\alpha$. The error bar represents the standard deviation. The thickness of the NLC film is 20 μm.

characterized by the vector $(0, \cos\alpha, \sin\alpha)$ (see inset of Fig. 4a). By tuning the angle $\alpha$ from 0° to 90°, the polarization is tuned from the $y$-direction to the $z$-direction while the form factor $\rho$ is tuned from 1 to 0. To quantify the tuning of the branched flow, we measure the scintillation index and SC for various polarization angles $\alpha$ (see Figs. 4d-e). The SC should be proportional to the square of the intensity of the extraordinary wave (Supplementary Note 3), thus it is proportional to $\rho^2 = \cos^4\alpha$. Indeed, the measured SC agrees well with a $\cos^4\alpha$ fit (Fig. 4e). These results demonstrate a continuous tuning of the branched flow by the polarization of the incident light and thus add a new dimension for the manipulation of the statistical wave dynamics, which is absent in other disordered systems[1,7,8,11,16,21].

## Discussion

In conclusion, we achieve here the experimental realization of the electrical tuning of the branched flow of light. Branched flow is a universal phenomenon emerging in the regime between ballistic and diffusive transport. While ballistic transport is featured with coherent wave propagation and diffusive transport with random and strong scattering, branched flow is featured with a mixture of randomness and wave behavior, yielding random caustics and erratic wave dynamics. Manipulation of branched flows empowers our control over the physical processes in this new regime which is of central importance [27,30,39]. The study here shows that the NLC system offers an ideal platform for such a purpose, thanks to the fluidic, anisotropic, and electro-optical properties of liquid crystals[31,32]. The recent advances of photo-alignment technology giving access to arbitrarily designed optical potential landscapes[40–44] offer more opportunities for liquid crystal systems in the manipulation of branched light flows, providing a promising future for the study of fundamental physics and potential applications in optics and photonics.

## Methods

### Device fabrication

The liquid crystal cell in the experiment is composed of an NLC film (E7, Jiangsu Hecheng Display Technology) sandwiched between two glass plates coated with ITO. The NLC film thickness is determined by spacers with a thickness of 10 or 20 μm. For disordered NLC samples, the glass plates are not specially treated and the infiltrated NLC forms schlieren textures by self-assembly. For the uniform NLC cell, we implement the conventional photo-alignment technique to fabricate the uniform-alignment liquid crystal[42]. The polarization-sensitive azo-dye SD1 is used as the alignment material. Under ultraviolet (UV) light exposure, the SD1 molecules tend to reorient their absorption oscillators perpendicular to the UV light polarization and further guide the liquid crystal directors. Thus, the aligned director along the $y$-direction is realized by simply controlling the polarization angle ($x$-direction) of illuminated UV light.

### Optical characterization

The spatial director distributions of the NLC film are measured by a birefringence imaging microscope (Crystalent-50, Ningcui). In order to generate and observe the branched flow of light, we have built two experimental configurations (Supplementary Fig. S8) to couple the quasi-plane-wave or focused Gaussian beam (laser wavelength of 532 nm, linewidth <0.01 pm, Changchun Laser Optoelectronics Technology Co., Ltd.) into the NLC film. The cylindrical lens and objective lens (10 × magnification) introduce a wide quasi-plane-wave beam and a focused Gaussian beam, respectively. The waist width of the Gaussian beam is approximately 20 μm, and the waist width of the quasi-plane-wave beam is several millimeters (Supplementary Fig. S10). The curved air/NLC interface is observed by infiltrating NLC into a silica glass tube (Supplementary Fig. S9), which can influence the light coupling. The

high-precision three-dimensional translation stage is implemented to tune the light coupling into the NLC film. The input polarized field (532 nm laser) is controlled by a polarizer and a half-wave plate. The microscope with 10 times objective lens is used to observe the branched flow of light in the NLC film. An optical camera (MIchrome 5 Pro, Shanghai Taizi Technology) is used to collect the intrinsic light scattering from NLC film.

## Simulations of electric-field-driven orientations of NLC

We simulate the liquid crystal orientations to the gating electrical field in the commercial software Tech Wiz-LCD-3D. In the simulations, in order to save computing resources, we use a simplified two-dimensional model since the gating electrical field is perpendicular to the liquid crystal film. The detailed simulation parameters are as follows: the liquid crystal (E7) film thickness 20 μm; the spray elastic constant ($K_{11}$) 11.09 pN, the twist elastic constant ($K_{22}$) 5.82 pN, the bend elastic constant ($K_{33}$) 15.97 pN, the rotational viscosity 34 mPa·s, the dielectric anisotropy 13.9; the liquid crystal molecules are set initially in-plane aligned and the upper and lower boundary layer of liquid crystal are set as strong anchoring boundary condition.

## Data availability

All data are available in the manuscript and the supplementary information. All related data are accessible in Figshare. Source data are provided in this paper.

## Code availability

We use the commercial software COMSOL MULTIPHYSICS to perform electromagnetic wave simulations and use the Tech Wiz-LCD-3D to perform the electric voltage response of NLC film. All related codes can be built using the instructions in the Method section.

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

## Acknowledgements

The authors thank Dr. Yi-feng Xiong, Prof. Xiaoshi Qian, and Prof. Yi Xu for their helpful discussions. Y.-q.L. thanks the support of the National Key Research and Development Program of China (No. 2022YFA1405000). This work was supported by the National Natural Science Foundation of China (Grants No. 12274357, 62005231 to J.-h.C., 12125504 to J.-H.J., 62075186 to L.J.C.), Natural Science Foundation of Fujian Province of China (2023J06011, 2022J01064 to J.-h.C.), and Fundamental Research Funds for the Central Universities (20720210045 to J.-h.C.). J.-h.C. also thanks the support of Xiaomi Young Talents Program/Xiaomi Foundation.

## Author contributions

S.-s.C. and K.-H.W. contributed equally to this work. S.-s.C. and K.-H.W. fabricated the samples with the assistance of S.-j.L., S.-j.G., P.C. and L.-J.C.; S.-s.C. performed the experiments with the assistance of K.-H.W. and S.-j.L.; S.-s.C. and J.-h.C. built the theoretical model with the assistance of J.-H.J., J.-b.W, W.H., Y.X., H.C. and D.H.; S.-s.C. and J.-h.C. analyzed the results and drew the figures with the help of Z.-K.L., J.-H.J. and D.-Q.Y.; J.-h.C. conceived the idea and co-supervised the project with J.-H.J. and Y.-q.L. All the authors contributed to the discussions of the results and the manuscript preparation. S.-s.C., P.C., J.-H.J., Y.-q.L. and J.-h.C. wrote the manuscript and the Supplementary Information.

## Competing interests

The authors declare no competing interests.
