## [Peer Review File · Nature Communications]

Electrical tuning of branched flow of lightREVIEWER COMMENTS

Reviewer #1 (Remarks to the Author):

This work reports an original approach to branched flow of light using a textured liquid crystal film. The experimental data and the modeling are interesting as such. The manuscript, however, call for several comments:

- Starting from the title, the study mentions the “control” of the phenomenon. Still, the reported results make me see it as a tuning/activation instead. Indeed, the potential depth can be reduced down to almost zero by electrical means without possibility to manipulate the spatial properties of the flow at fixed irradiation for a given statistical signature (i.e., for a given potential depth). Such a tuning could also be reached by temperature control as the birefringence vanishes as the temperature is getting closer to the melting point, the which would have the virtue of preserving the director field structure, especially noting that this is an issue in the present work.

- The disordered potential is claimed to be smooth, which is not correct since the Schlieren structure is described as a random distribution of singular defects.

- Waveguiding effect are said negligible, however the e-field can be expectedly trapped since the refractive index of usual glass is smaller than the extraordinary refractive index, this deserves clarification.

- The medium properties varies in (x,y) plane due to space varying director field. If e-waveguiding happens, it is also space varying, which leads to space varying nondissipative propagation loss as light propagates. This would deserve some comments in the continuation of the previous comment.

- The input beam waist value is seemingly missing. The beam injection detail (air/LC curved interface or glass/LC) would also be welcome.

- p7 the branching flow is said to be self-similar, which is not supported experimentally.

- p7, the almost pure o-field at large voltage is said “collimated” and then “focused and collimated”, while one could expect merely propagating and experiencing natural diffraction leading to increasing width and decreasing on-axis intensity.

- p7, conclusions are made on o-wave behavior from electric field induced behavior. Here should be shown o-wave comparative data to support the discussion.

- Comparison with homogeneous unperturbed cell with director along y would allow clarifying the effects of non-smooth nature of the director field for both o and e waves.

- Rayleigh distribution is mentioned without presentation of what it is.

- p8, Schlieren texture is said to be modified, which is far too vague regarding Fig.S4 that shows structural changes as voltage is applied.

- Evolution of the fidelity parameter in homogeneous system would be good analytical benchmark to superimposed to the data. Experimentally as well by using homogeneous benchmark LC sample.

- “quasi-plane-wave” wording is questionable due focused beam and possible curved injection interface.

- It is unclear what is quantitatively made from the introduction of the quantity rho since only plot of the formula is shown (by the way, 2D plot Fig.3e would be enough, if eventually useful to show)

- Method: why curing the glue with UV that would define a fixing spacing would be a problem for the bare glass part of the substrate that will be in contact with LC after filling since UV lamps are usually unpolarized?

- p11, simulations are mentioned: it is unclear whether Schlieren texture is simulated or not.

- Normalizing the applied voltage difference to the Fréedericksz threshold voltage would help clarifying the electrical reorientation overall picture across the paper.

- Fig.S11 : arrows and segments are barely readable

- Minor points: n_{\perp} and n_{\parallel} would be clearer and more usual than n_1 and n_2 . The nematic director field is usually written as n instead of m . The director should be defined as being local average molecular orientation, and, subsequentlys “averaged director” wording is unclear. Thickness to be given when experiment is presented.

- From the reading point of view: English should be improved and the text appears repetitive at several places.

- In my opinion, the structure of the paper is making the overall reading lacks clarity and fluidity.

In summary, I recommend to question and reconsider the way the results are reported.

Reviewer #2 (Remarks to the Author):

Key results

This manuscript studies the branched flow of light in a nematic liquid crystal (NLC). The crystal is sandwiched between two glass plates. An electric potential can be applied through the resulting cell. The birefringent properties are changed by applying a voltage in the range of 5 to 10 V. The inhomogeneities in the NLC are sufficient to trigger the formation of a branched flow pattern. This is related to the extraordinary refractive index of the NLC and is completely suppressed by the applied voltage. The process is reversible, for the inhomogeneities are frozen in the cell and not significantly affected by the external potential.

Validity

The results are sound and valid.

Significance

For branched flows occur in many different physical settings, this approach to their suppression is very significant and timely, after the first observation of a branched flow of light in 2020 (Ref. 13 of the present manuscript).

Data and methodology

The use of different beam profiles (a narrow beam and a broad beam-nearly plane wave), the assessment of polarization effects allow the authors to validate the proposed explanation of the observed phenomena.

There is though a gap in the voltage values applied to the cell. In Fig. 2, V_{pp} jumps from 6 to 10 V and in Fig. 3 from 5 to 10 V. From panel 3(g) it looks like there could be more to study in this gap. Please comment.

Moreover, V_{pp} is never explicitly correlated to the parameters of the effective potential for light propagation, Eq. (2) or (S20). From the discussion, I infer that the correlation length is not affected, but the variance of the potential changes. Such an analysis could improve the quantitative nature of this work and better identify the threshold above which the branching distance becomes much larger than the cell size.

Figs. S3, S6, and S7 are not discussed in the Supplemental text. What is their role? I think these results should be discussed, if their presentation is deemed essential, or else removed.

Analytical approach

The analytical and numerical approaches are confined mainly to Supplemental Material. The techniques are standard and well understood. I appreciate the use of the probability

distribution in Fig. 2 (b-c) to show the extreme and rare nature of caustics (the ingredient leading to branching) and how they disappear for a large enough voltage. I was wondering if a similar distribution could be presented for the plane wave in Fig. 3.

The derivations reported in Section S4 should be clarified: in its present form its role in the interpretation of data is not fully justified. Notice that $m(x)$ is used here and could be confusing, since \mathbf{m} is used to denote the director.

Suggested improvements

Panels showing light intensity distribution are quite small in the text and axes are not labelled. In black&white, the small white fonts and the scale bars are not readable. I suggest to try another graphical representation.

As written above, I would stress more on the quantitative correspondence between $V^{\text{sup}}_{\text{pp}}$ and the branching length parameters.

I suggest that the authors check that every acronym is consistent and every quantity is defined in their formulas.

Final consideration

My expertise on branched flows is quite limited. I would suggest to expand the introductory paragraph, adding one or two sentences for non experts. The reference list is quite broad, I think most of the relevant papers were appropriately cited.

Response Letter

We are very grateful for the reviewers' recognition of the innovation and quality of our work. We also appreciate the reviewers' comments and suggestions, which are helpful for improving our manuscript. The main text and the supplementary information are revised carefully according to these comments and suggestions (All revisions are labeled in colors). In the following, please find the detailed point-by-point responses.

Reply to Reviewer #1

Reviewer's remark: *This work reports an original approach to branched flow of light using a textured liquid crystal film. The experimental data and the modeling are interesting as such.*

Our reply: We thank the reviewer for his/her appreciation of our work. We have taken consideration of the reviewer's comments and suggestions thoroughly, according to which we revise the manuscript carefully.

Reviewer's comments: *Q1: Starting from the title, the study mentions the "control" of the phenomenon. Still, the reported results make me see it as a tuning/activation instead. Indeed, the potential depth can be reduced down to almost zero by electrical means without possibility to manipulate the spatial properties of the flow at fixed irradiation for a given statistical signature (i.e., for a given potential depth). Such a tuning could also be reached by temperature control as the birefringence vanishes as the temperature is getting closer to the melting point, the which would have the virtue of preserving the director field structure, especially noting that this is an issue in the present work.*

Our reply: We admit that the word "control" is a bit over. We switch to the new title: *"Electrical tuning of branched flow of light"*.

Indeed, thermal tuning is another useful technique for the manipulation of photons in liquid crystals. We agree with the reviewer that using temperature control can also tune the branched flow of light as shown in Fig. R1a. Nevertheless, we find that the thermal tuning method is highly irreversible by comparing the cross-section field distribution of Fig. R1b and Fig. R1c for the heating and cooling processes. This is probably due to the collision of the growing isotropic-nematic interface during the cooling process, which can result in a higher energy metastable state [Science Advances, 4(11), eaau8064.]. In contrast, the electrical tuning method can realize highly reversible manipulations of the branched patterns since the topological structures of liquid crystals are stable under such tuning (if the electrical field is not too high). We note that as suggested by Reviewer #2, all the optical field landscape plots have been changed with the as-captured images for clarity.

In the revised manuscript, we have added extra discussions on the thermal tuning of the branched flow of light: *"We note that the branched flow field patterns can also be tuned by temperature control as*

the birefringence vanishes when the temperature is approaching the phase transition point of NLC (Supplementary Fig. S20a). However, we observe that the thermal tuning method is highly irreversible for the branched photonic patterns (Supplementary Fig. S20b-c). This is probably due to the collision of the growing isotropic-nematic interface during the cooling process, which can result in a higher energy metastable state of the NLC and complex changes in the schlieren textures³⁶.” Figure R1 is added to the Fig. S20 of Supplementary Materials with extra discussions.

Fig. R1 Temperature tuning of the branched flow of light. (a) Thermal switch off and on of the branched flow of light upon temperature cycling. The scale bar is 100 μm . (b-c) The single-shot traces of cross-section optical intensity at the white-dashed line in (a) during the processes of heating (b) and cooling (c).

Reviewer's comments: Q2: The disordered potential is claimed to be smooth, which is not correct since the Schlieren structure is described as a random distribution of singular defects.

Our reply: We agree that the disordered potential is not smooth and there is actually no need to require it to be smooth for the emergence of the branched flow of light. In the revised manuscript, we corrected the description of the disordered potential as follows: *“Such disordered orientations lead to inhomogeneous dielectric anisotropy in the NLC film which acts as a weakly disordered potential for the propagating light”*.

Reviewer’s comments: Q3: *Waveguiding effect are said negligible, however the e-field can be expectedly trapped since the refractive index of usual glass is smaller than the extraordinary refractive index, this deserves clarification.*

Our reply: Indeed, the previous statement is a bit unclear. We agree with the reviewer that the both *o*-wave and *e*-wave are confined in the planar waveguide structure (Fig. R2a), since the refractive index of nematic liquid crystal (E7) is larger than that of glass cladding. Our initial purpose is to say that the confinement effect for the fundamental modes does not modify the effective refractive index considerably. As shown in Fig. R2b, the effective refractive index of the fundamental mode is very close to the refractive index of the liquid crystal. This is understandable because the thickness of liquid crystal film ($\geq 10 \mu\text{m}$) is much larger than the light wavelength (532 nm). In this regard, the waveguiding-induced effective index change can be neglected in the optical potential.

In the revised manuscript, we have added extra discussions on the waveguide effect: *“Here, since the NLC film’s thickness is much larger than the wavelength of the light, the effective index of the waveguide mode is very close to the liquid crystal index (Supplementary Fig. S3), while the waveguiding-induced effective index change can be neglected.”* And Fig. R2 is added to the Supplementary Fig. S3 in Supplementary Materials with extra discussions.

Fig. R2 (a) Schematic of planar waveguide structure for modeling light field in liquid crystal cell. (b) Calculated effective index of the fundamental mode of light depending on the thickness of liquid crystal film. Here we consider the *o*-wave of liquid crystal with refractive index $n_2=1.52$, and the glass cladding index $n_1=1.5$. The cladding thickness is assumed to be infinite since the practical glass thickness is ~ 1.1 mm, which is far larger than the core thickness.

Reviewer’s comments: Q4: *The medium properties vary in (x, y) plane due to space varying director field. If e-waveguiding happens, it is also space varying, which leads to space varying nondissipative*

propagation loss as light propagates. This would deserve some comments in the continuation of the previous comment.

Our reply: We agree that the space-varying dielectric tensor affects both the in-plane light scattering and the nondissipative propagation loss. As the light scattering by the space-varying refractive index is rather weak, we are not sure whether such light scattering is the main mechanism of the nondissipative propagation loss. There are other mechanisms such as light scattering due to long-range collective orientation fluctuations of the molecular axis of liquid crystal [Molecular Crystals, 7, 325-345 (1969)] and the roughness of the interfaces between the NLC and the cladding materials, which could be more prominent.

On the other hand, the nondissipative propagation loss is necessary for our experiments: We detect the light field distribution via the out-of-plane radiation that comes from such nondissipative propagation loss. As we did not observe notable changes in the out-of-plane radiation intensity during electrical tuning of the branched flow of light or by changing the light polarization. We believe that the space-varying refractive index may not be the main reason for the nondissipative propagation loss.

In the revised manuscript, we have added extra discussions on the loss issue: *“Due to the considerable non-dissipative propagation loss from the disordered director distribution of NLC film, for example, light scattering due to long-range collective orientation fluctuations of molecular axis of NLC³⁷ ...”*

Reviewer’s comments: *Q5: The input beam waist value is seemingly missing. The beam injection detail (air/LC curved interface or glass/LC) would also be welcome.*

Our reply: In the experiments, we implemented two kinds of optical beams as input fields, i.e., focused Gaussian beam and broad quasi-plane wave beam. The waist width of the Gaussian beam is approximately 20 μm , and the waist width of the quasi-plane wave beam is several millimeters. The curved air/LC (E7) interface in the cell structure with gap of 10-20 μm is challenging to observe. Here we measure the air/LC interface by infiltrating LC into a silica glass tube as shown in Fig. R3. The contact angle of $\sim 37^\circ$ between glass and LC is determined by their surface tension coefficient and interface interaction (The Journal of Adhesion, 17, 123-134 (1984)). The curved air/NLC interface can influence the light coupling.

In the revised manuscript, we have added extra descriptions on input beam parameters in Methods section: *“The waist width of the Gaussian beam is approximately 20 μm , and the waist width of the quasi-plane-wave beam is of several millimeters (Supplementary Fig. S10)”*; and the beam injection detail in Methods section: *“The curved air/NLC interface is observed by infiltrating NLC into a silica glass tube (Supplementary Fig. S9), which can influence the light coupling.”* Figure R3 is added to the Fig. S9 of Supplementary Materials with extra discussions.

Fig. R3 Microscope image of liquid crystal infiltrated in a silica glass tube. The red dashed line labels the curved air-liquid-crystal-glass interface.

Reviewer's comments: Q6: p7 the branching flow is said to be self-similar, which is not supported experimentally.

Our reply: It is indeed a problem that the previous images of the optical field do not show clearly the self-similar branching behaviours. In Fig. R4, we show that the self-similar branching geometry can be clearly seen. The self-similar branching geometry indicates that the branch can continue to split into smaller tributaries after the first branch. In the revised manuscript, we have added Fig. R4 to Fig. 1e with extra descriptions: “...the landscapes of branched flow of light can be directly observed by an optical microscope system with a digital camera. The propagating light continues to branch off into even smaller tributaries, showing an intriguing, complex, self-similar branching geometry (see Fig. 1e)”.

Fig. R4 Observations of cascaded branching geometry. The white dashed box shows the self-similar features. The scale bar is 200 μm .

Reviewer's comments: Q7: p7, the almost pure o-field at large voltage is said “collimated” and then “focused and collimated”, while one could expect merely propagating and experiencing natural diffraction leading to increasing width and decreasing on-axis intensity.

Our reply: We thank for the reviewer's helpful comment. We have revised these descriptions in the revised manuscript:

“...leading to a beam propagation experiencing natural diffraction with increasing width and decreasing on-axis intensity”.

“...the light beam simply experiences natural diffraction...”

Reviewer's comments: Q8: p7, conclusions are made on o-wave behavior from electric field induced behavior. Here should be shown o-wave comparative data to support the discussion.

Our reply: We measure the optical landscapes in a disordered liquid crystal cell for different polarization inputs, as shown in Fig. R5. The e-wave excited fields manifest typical branched-flow geometry, on the contrary, the o-wave excited fields show natural diffraction features. Note that there exist minor branched structures even for o-wave excitations probably due to the small pretilt angle of liquid crystals in the interface of the glass substrate. In addition, we have also discussed the polarization effect for quasi-plane-wave input in Fig. 4 of the main text, which supports the discussion.

We have added Fig. R5 to the Fig. S18 of Supplementary Materials, and supplemented discussions on the issue in the main text: *“In fact, the branched flow pattern can be tuned by the polarization of the incident light (Supplementary Fig. S18), as shown and discussed later.”*

Fig. R5 Optical field landscapes in disordered liquid crystal cell for (a) e-wave excitation (in-plane polarization), (b) o-wave excitation (out-of-plane polarization). The scale bar is 200 μm .

Reviewer's comments: Q9: Comparison with homogeneous unperturbed cell with director along y would allow clarifying the effects of non-smooth nature of the director field for both o and e waves.

Our reply: Thanks for the suggestion! We implement the conventional photoalignment technique to fabricate the uniform alignment liquid crystal (*Advanced Materials* **26**, 1590-1595 (2014)). The polarization-sensitive azo-dye SD1 is used as the alignment material. Under UV exposure, the SD1 molecules tend to reorient their absorption oscillators perpendicular to the UV light polarization and further guide the liquid crystal directors. Thus, the aligned director along the y-direction is realized by simply controlling the polarization angle (x-direction) of illuminated UV light. For the homogeneously aligned liquid crystal film, neither branched patterns nor caustics can be observed for both the quasi-plane wave input and Gaussian beam input, as shown in Fig. R6. In addition, the propagating fields show similar landscapes without branching structures for the light polarization along the in-plane (y-direction) and out-of-plane (z-direction). These experimental results unambiguously clarify the effects of inhomogeneous director distributions for branched flow generation.

Fig. R6 Optical field landscapes for quasi-plane wave input of light polarization (a) along y-direction, and (b) along z-direction. Propagating field landscapes for Gaussian beam input with input polarization (c) along y-direction, (d) z-direction. The scale bar is 200 μm .

In the revised manuscript, we have added the discussions on homogeneous cell: *“We also fabricate a uniform-alignment NLC cell (see Methods) and the propagating light only shows natural diffraction features (Supplementary Fig. S11).”* We also added fabrications of uniform liquid crystal cells in the Methods: *“For the uniform NLC cell, we implement the conventional photoalignment technique to fabricate the uniform-alignment liquid crystal⁴². The polarization-sensitive azo-dye SD1 is used as the alignment material. Under UV exposure, the SD1 molecules tend to reorient their absorption oscillators perpendicular to the UV light polarization and further guide the liquid crystal directors. Thus, the aligned director along y-direction is realized by simply controlling the polarization angle (x-direction) of illuminated UV light.”* And we have included Fig. R6 as Fig. S11 in the revised Supplementary Materials with extra discussions.

Reviewer’s comments: Q10: Rayleigh distribution is mentioned without presentation of what it is.

Our reply: The Rayleigh statistics feature a negative-exponential intensity probability density function $P_{\text{Rayleigh}}(I) = e^{-I}$, where $I \sim |E|^2$ is the normalized field intensity [Phys. Rev. Lett. 104, 093901 (2010)]. We have added extra discussions on this issue in the main text: *“In conventional light scattering, such a distribution will be a Rayleigh distribution: $P(I) = e^{-I}$, where I is the normalized field intensity⁷.”*

Reviewer’s comments: Q11: p8, Schlieren texture is said to be modified, which is far too vague regarding Fig.S4 that shows structural changes as voltage is applied.

Our reply: We have updated the supplementary figure for clarity, as shown in Fig. R7 (Note that the previous figure covers a too large region. Here, the updated one gives a smaller region where the schlieren texture can be seen clearly). It can be found that the gate voltages only modify the schlieren texture without destroying the mesoscopic structures which are fixed by some unknown reason (possibly the

surface roughness and interactions or the stability of topological defects), and consequently, the branched flows are highly reversible with the electrical voltage tuning.

Fig. R7 (a, b) Polarized microscope image of disordered liquid crystal film (a) with the increase of gate voltages and (b) with the decrease of gate voltages. The scale bar is 100 μm . The arrows indicate the polarizer and analyzer directions.

Reviewer's comments: Q12: *Evolution of the fidelity parameter in homogeneous system would be good analytical benchmark to superimposed to the data. Experimentally as well by using homogeneous benchmark LC sample.*

Our reply: We have conducted experiments in a uniform cell (fabricated by photoalignment technique discussed in Q9) and measured the corresponding propagation fields as shown in Fig. R8. The extracted evolution of the fidelity parameter for Gaussian beam input is shown in the blue curve of Fig. R8a. The homogeneous sample shows a similar fidelity degradation curve compared with that of the disordered sample under a high gate voltage of 10 V. As for the quasi-plane-wave input in the homogeneous sample, the evolving fidelity curve agrees well with the results of high-gating-voltage (10 V) disordered sample (Fig. R8b). These complementary experiments further clarify the origin of electrical field tuning branched flow.

We have updated Fig. 2f and Fig. 3f with Fig. R8a and Fig. R8b, respectively in the revised main text and added extra discussions on this issue:

“We also fabricate a uniform-alignment NLC cell (see Methods) and the propagating light only shows natural diffraction features (Supplementary Fig. S11). We measure the fidelity curve of propagating fields as a benchmark (blue curve in Fig. 2f), which shows a similar decay rule compared to the high gating-voltage ($V_{pp}=10\text{ V}$) disordered sample (yellow curve in Fig. 2f).”

“The fidelity of propagating fields in the uniform NLC cell is nearly the same as that of the high gating-voltage ($V_{pp}=10\text{ V}$) disordered sample, as expected.”

Fig. R8 Measured optical field fidelity versus the propagation distance along the x -direction in disordered liquid crystal cell and homogeneous cell (uniform alignment) with (a) Gaussian beam and (b) quasi-plane wave. The fidelity of propagating fields in a uniform liquid crystal cell is used as a benchmark

Reviewer’s comments: Q13: “quasi-plane-wave” wording is questionable due focused beam and possible curved injection interface.

Our reply: We measure the optical field evolutions in a homogeneous alignment liquid crystal cell, as shown in Fig. R9. These nearly uniform fields indicate that the curved injection interface has a negligible effect on the propagating fields. In the revised manuscript, we have included Fig. R9 as Fig. S10 in the revised Supplementary Materials with extra discussions.

Fig. R9 (a) The light field in a homogeneous unperturbed cell with director along y coupled by a quasi-plane wave with polarization along y -direction. (b) The cross-section intensity distribution at the dashed line position of (a). The scale bar is $100\ \mu\text{m}$.

Reviewer's comments: Q14: It is unclear what is quantitatively made from the introduction of the quantity ρ since only plot of the formula is shown (by the way, 2D plot Fig.3e would be enough, if eventually useful to show)

Our reply: The local formation factor ρ quantifies the portion of the e-wave excitation for the linear-polarization incident light, as shown in Fig. R10. Only the e-wave light in the liquid crystal film experiences the non-uniform effective refractive index and forms the branched flow. We thus use the ρ quantity to conduct numerical calculations and then compare such calculations with the experimental results. In fact, we have used this approach in the numerical calculations of electrical tuning (Fig. 3d in the main text) and polarization tuning (Fig. 4e in the main text) of the branched flow of light in our system.

In the revised main text, we have added extra discussions on this issue: *“The electric voltage induced change of the ρ -factor can be used to quantitatively calculate the statistical properties of branched flow fields.”*, and we have updated Fig. 3e with the 2D plot of Fig. R10.

Fig. R10 Dependence of the formation factor ρ on the azimuth angle φ and the polar angle θ of the liquid crystal director. Inset: Schematic of the NLC director \hat{n} and the optical wavevector \hat{k} along the x -direction

Reviewer's comments: Q15: Method: why curing the glue with UV that would define a fixing spacing would be a problem for the bare glass part of the substrate that will be in contact with LC after filling since UV lamps are usually unpolarized?

Our reply: The unpolarized ultraviolet lamp certainly will not affect the bare glass substrate. We have deleted these descriptions in the revised manuscript.

Reviewer's comments: Q16: p11, simulations are mentioned: it is unclear whether Schlieren texture is simulated or not.

Our reply: We simulate the molecules' reorientation in the liquid crystal due to the gate voltage using the commercial software Tech Wiz-LCD-3D. In these simulations, in order to save computing resources,

we use a simplified two-dimensional model since the gating electric field is perpendicular to the liquid crystal film. The detailed simulation parameters are as follows: the liquid crystal (E7) film thickness 20 μm ; the splay elastic constant (K_{11}) 11.09 pN, the twist elastic constant (K_{22}) 5.82 pN, the bend elastic constant (K_{33}) 15.97 pN, the rotational viscosity 34 mPa·s, the dielectric anisotropy 13.9; the liquid crystal molecules are set initially in-plane aligned and the upper and lower boundary layer of liquid crystal are set as strong anchoring boundary condition. From the Fig. 3d in the main text, it shows that the simulated results based on the simplified model agree reasonably well with the experimental results.

In the revised main text, we have added these simulation details in the Methods section.

Reviewer's comments: Q17: Normalizing the applied voltage difference to the Fréedericksz threshold voltage would help clarifying the electrical reorientation overall picture across the paper.

Our reply: Thanks for the helpful comments! The Fréedericksz threshold voltage V_c can be calculated via the following formula [Yang D.K., Wu S.T. Fundamentals of liquid crystal devices. John Wiley & Sons, 2014] :

$$V_c = \pi \sqrt{\frac{K_{11}}{\epsilon_0 \Delta\epsilon}}$$

where the splay elastic constant is $K_{11}=11.09$ pN, ϵ_0 is vacuum dielectric constant, and the dielectric anisotropy $\Delta\epsilon = \epsilon_{\parallel} - \epsilon_{\perp} = 13.9$ can be obtained for E7 material [Liquid Crystals, 46, 349-355 (2019)]. Thus, the calculated Fréedericksz threshold voltage is about $1.8 V_{pp}$, which is close to the experimental results ($\approx 1.6 V_{pp}$).

In the revised main text, we have added extra discussions on the Fréedericksz threshold: *“Moreover, we find that the observed transition voltage is close to the calculated Fréedericksz transition threshold $\approx 1.8 V$ (Supplementary Section 3), which clarifies the electrical reorientation of the liquid crystal molecules as the underlying mechanism for the electrical tuning of the branched flow.”* The explicit calculation of the Fréedericksz threshold is added to the revised Supplementary Materials.

Reviewer's comments: Q18: Fig. S11: arrows and segments are barely readable

Our reply: We have replotted the supplementary figure as shown in Fig. R11.

FIG. R11. Cross-sectional view of NLC director when the driving voltage (V_{pp}) is (a) 0 V and (b) 5 V.

Reviewer's comments: Q19: *Minor points: n_{\perp} and n_{\parallel} would be clearer and more usual than n_1 and n_2 . The nematic director field is usually written as n instead of m . The director should be defined as being local average molecular orientation, and, subsequently “averaged director” wording is unclear. Thickness to be given when experiment is presented.*

Our reply: We appreciate the reviewer for the nice suggestions. We have changed the symbols of the birefringence index and the director field. The definition of director is changed to the local average molecular orientation: *“This unit vector determines the local average molecular orientation.”* The average director is a global average director field of liquid crystal in a certain measurement region. To avoid misunderstanding, we have deleted the related descriptions of “averaged director” in the revised main text. We have added descriptions of cell thickness in captions of Figures 1-4.

Reviewer's comments: Q20: *From the reading point of view: English should be improved and the text appears repetitive at several places. In my opinion, the structure of the paper is making the overall reading lacks clarity and fluidity. In summary, I recommend to question and reconsider the way the results are reported.*

Our reply: In the revised manuscript, we have polished the English writings and have improved the clarity and fluidity of the manuscript.

Reply to Reviewer #2

Reviewer's remarks: **Key results:** *This manuscript studies the branched flow of light in a nematic liquid crystal (NLC). The crystal is sandwiched between two glass plates. An electric potential can be applied through the resulting cell. The birefringent properties are changed by applying a voltage in the range of 5 to 10 V. The inhomogeneities in the NLC are sufficient to trigger the formation of a branched flow pattern. This is related to the extraordinary refractive index of the NLC and is completely suppressed by*

the applied voltage. The process is reversible, for the inhomogeneities are frozen in the cell and not significantly affected by the external potential. **Validity:** The results are sound and valid. **Significance:** For branched flows occur in many different physical settings, this approach to their suppression is very significant and timely, after the first observation of a branched flow of light in 2020 (Ref. 13 of the present manuscript).

Our reply: We thank the reviewer for his/her appreciation of our work. We have taken consideration of the reviewer's comments and suggestions thoroughly, according to which we revise the manuscript carefully.

Reviewer's comments: Data and methodology: Q1: The use of different beam profiles (a narrow beam and a broad beam-nearly plane wave), the assessment of polarization effects allow the authors to validate the proposed explanation of the observed phenomena.

Our reply: For the narrow beam excitations, we measure the optical landscapes in a disordered liquid crystal cell for different polarization inputs, as shown in Fig. R12. The e-wave excited fields manifest typical branched-flow geometry, in contrast, the o-wave excited fields show natural diffraction features with increasing width and decreasing on-axis intensity. Note that there exist minor branched structures even for o-wave excitations probably due to the small pretilt angle of liquid crystals in the interface of the glass substrate. For broad beam-nearly plane wave excitations, we have discussed the polarization effect for quasi-plane-wave input in Fig. 4 of the main text. These experimental results clearly clarify the proposed explanation of observed phenomena, i.e., extraordinary wave dominates the branched flow.

We have added Fig. R12 to the Fig. S18 of Supplementary Materials, and supplemented discussions on the issue in the main text: *“In fact, the branched flow pattern can be tuned by the polarization of the incident light (Supplementary Fig. S18), as shown and discussed later.”*

Fig. R12 Optical field landscapes in disordered liquid crystal cell for (a) e-wave excitation (in-plane polarization), (b) o-wave excitation (out-of-plane polarization). The scale bar is 200 μm . From (b) one can infer that there are events where part of the o-wave is scattered into e-wave.

Reviewer's comments: Q2: There is though a gap in the voltage values applied to the cell. In Fig. 2, V_{pp} jumps from 6 to 10 V and in Fig. 3 from 5 to 10 V. From panel 3(g) it looks like there could be more to study in this gap. Please comment.

Our reply: In the previous figures, we plot fewer voltage-data curves for clear illustrations, since the electrical voltages induced orientations of liquid crystal are close to saturations when the voltage is larger than 5 V, as shown in Fig. 3d in the main text. In Fig. R13a, we have added an extra voltage curve (8 V), which shows saturated features compared with that of 10 V. In Fig. R13 (b-c), we have added extra voltage curves (7 V, 9V), which show saturated features compared with that of 10 V. In addition, as suggested by Reviewer 1, we have included the performance of a uniform liquid crystal cell as a benchmark.

We have updated Fig. 2f, Fig. 3f, and Fig. 3g with Fig. R13a, Fig. R13b, Fig. R13c, respectively with extra discussions in the revised main text:

“We also fabricate a uniform-alignment NLC cell (see Methods) and the propagating light only shows natural diffraction features (Supplementary Fig. S11). We measure the fidelity curve of propagating fields as a benchmark (blue curve in Fig. 2f), which shows a similar decay rule compared to the high gating-voltage ($V_{pp}=10$ V) disordered sample (yellow curve in Fig. 2f).”

“The fidelity of propagating fields in the uniform NLC cell is nearly the same as that of the high gating-voltage ($V_{pp}=10$ V) disordered sample, as expected.”

Fig. R13 Measured optical field fidelity versus the propagation distance along the x -direction in disordered liquid crystal cell and homogeneous cell (uniform alignment) with (a) Gaussian beam and (b) quasi-plane wave. The fidelity of propagating fields in a uniform liquid crystal cell is used as a benchmark. (c) Measured average branched density versus the propagation distance for various electric voltages.

Reviewer’s comments: Q3: Moreover, V_{pp} is never explicitly correlated to the parameters of the effective potential for light propagation, Eq. (2) or (S20). From the discussion, I infer that the correlation length is not affected, but the variance of the potential changes. Such an analysis could improve the quantitative nature of this work and better identify the threshold above which the branching distance becomes much larger than the cell size.

Our reply: The reviewer is right that in our platform, the gate voltages mainly manipulate the potential strength by controlling the out-of-plane orientation angle (θ) of the liquid crystal director, and the correlation length is almost not affected as shown in Fig. R14. Although the branching distance can be

changed by the gate voltages, it is, however, difficult to directly observe this effect in reality, since the inevitable excitations of ordinary-wave fields can mix with the branched fields of the extraordinary wave.

In the revised main text, we have added extra discussions on this issue: *“We note that the gate voltages mainly manipulate the optical potential strength by controlling the out-of-plane orientation angle of liquid crystal director, whereas the correlation length of the optical potential is not considerably affected (Supplementary Fig. S15). Although the branching distance can be modified by the gate voltage, it is very difficult to directly observe this effect because of the considerable loss in light propagation and the inevitable excitation of the ordinary-wave component.”* We have added Fig. R14 to Fig. S15 of the revised Supplementary Material with extra discussions.

Fig. R14 The first branching position, potential strength ϵ and correlation length l_c versus the polar angle of director.

Reviewer’s comments: Q4: Figs. S3, S6, and S7 are not discussed in the Supplemental text. What is their role? I think these results should be discussed, if their presentation is deemed essential, or else removed.

Our reply: We thank for the reviewer’s helpful comment and suggestion. Figure S3 (updated Fig. S6) discusses the influence of dielectric loss of liquid crystal on the evolving optical fields, which corresponds to the practical sample. Figure S6 (updated Fig. S12) shows the experimental results for liquid crystal cell of thickness 10 μm and 20 μm , and they exhibit similar branched flow features, which help clarify that the waveguiding effect can be negligible in our platform. Figure S7 (updated Fig. S13) shows the evolving optical fields with sequential gate voltages, which can be supplementary to the results in Fig. 3(a-d) of the main text.

In the revised Supplementary Materials, we have added extra descriptions of these figures for clarity.

Reviewer’s comments: Analytical approach: Q5: The analytical and numerical approaches are confined mainly to Supplemental Material. The techniques are standard and well understood. I appreciate the use of the probability distribution in Fig. 2 (b-c) to show the extreme and rare nature of caustics (the ingredient leading to branching) and how they disappear for a large enough voltage. I was wondering if a similar distribution could be presented for the plane wave in Fig. 3.

Our reply: We thank for the reviewer’s helpful comment and suggestion. Under the condition of quasi-plane wave coupling, we calculated the probability distribution of light intensity upon 0 V and 10 V as shown in Fig. R15. These statistical characteristics are similar to those of Gaussian beam coupling.

We have added Fig. R15 to the Fig. S14 in the revised Supplementary Materials with extra discussions.

Fig. R15 Measured profiles of the optical field with the quasi-plane incident for different electric voltages: (a) 0 V and (b) 10 V. The scale bar is 100 μm . (c) Statistical distribution of the cross-section intensity at the dashed line in (a). The inset shows the cumulative distribution function (CDF) of the cross-section intensity in the semi-log axis. The black curve represents the CDF of Rayleigh distribution. (d) Probability distributions of the optical intensity with an electric voltage bias of 10 V at the incident (olive green columns) and at the white-dashed line in (b) (greyish green columns). Inset: CDF of the two distributions of the optical intensity

Reviewer’s comments: Q6: The derivations reported in Section S4 should be clarified: in its present form its role in the interpretation of data is not fully justified. Notice that $m(x)$ is used here and could be confusing, since \mathbf{m} is used to denote the director.

Our reply: Actually, the explicit derivations of branched density have been reported in the theoretical work (*Phys. Rev. Lett.* **105**, 020601 (2010)). The current work mainly focuses on the experimental results. In the revised Supplementary Material, we add more detailed derivations of the important equations and

have added extra discussions and interpretations of the data. In addition, according to the suggestion of reviewer 1, the director symbol is changed to n for clarity.

Reviewer's comments: *Suggested improvements: Q7: Panels showing light intensity distribution are quite small in the text and axes are not labelled. In black & white, the small white fonts and the scale bars are not readable. I suggest to try another graphical representation.*

Our reply: Thanks for the suggestion. We revised the figures accordingly.

Reviewer's comments: *Q8: As written above, I would stress more on the quantitative correspondence between V_{pp} and the branching length parameters.*

Our reply: Although the branching distance can be modified by the gate voltage (as shown in Fig. R14), it is very difficult to directly observe this effect because of the considerable loss in light propagation and the inevitable excitation of the ordinary-wave component. As shown in Fig. 3c in the main text, the measured scintillation index curve cannot resolve a clear change of branching length via gate voltages. The quantitative correspondence between gate voltages and the (ideal) branching length can be calculated by the simulated liquid crystal orientations and the statistical rules (Fig. R14) of branched flow.

In the revised main text, we have added extra discussions on this issue: *“We note that the gate voltages mainly manipulate the potential strength by controlling the out-of-plane orientation angle of liquid crystal director, and the correlation length is not significantly affected (Supplementary Fig. S15). Although the branching distance can be modified by the gate voltage, it is very difficult to directly observe this effect because of the considerable loss in light propagation and the inevitable excitation of the ordinary-wave component.”* We have added Fig. R14 to the Fig. S15 in the revised Supplementary Materials with extra discussions.

Reviewer's comments: *Q9: I suggest that the authors check that every acronym is consistent and every quantity is defined in their formulas.*

Our reply: Thanks for the suggestion! We have checked carefully every acronym and every quantity for consistency and accuracy.

Reviewer's comments: *Final consideration: Q10: My expertise on branched flows is quite limited. I would suggest to expand the introductory paragraph, adding one or two sentences for non-experts. The reference list is quite broad, I think most of the relevant papers were appropriately cited.*

Our reply: In the revised main text, we added extra descriptions of branched flow in the introductory paragraph: *“In nonlinear wave dynamics, branched flows can also act as an activation mechanism for the appearance of extreme wave events such as tsunamis waves and rogue waves^{11,13-15}.”* In addition, Figure 1e is added to explicitly demonstrate the branched flow phenomenon.

REVIEWERS' COMMENTS

Reviewer #1 (Remarks to the Author):

The authors have convincingly updated the manuscript on many points. All comments raised in the first report have been taken into account. In my opinion, the revised version of the manuscript provides a thorough study of branched light flows in liquid crystals and is worthy of publication. This work illustrates well how liquid crystals, as optical materials, continue to be a platform for studying fundamental optical phenomena.

Reviewer #2 (Remarks to the Author):

The authors answered extensively to reviewers' comments.

Considering the other reviewer's comments and how the authors responded, I suggest publishing the article as it is.